# Social learning dynamics influence performance and career self-efficacy in career-oriented educational virtual environments

**Bradley D. Pitcher** [1], **Daniel M. Ravid**[2], **Peter J. Mancarella**[1], **Tara S. Behrend**[1] *

**1** Department of Psychological Sciences, Purdue University, West Lafayette, IN, United States of America,
**2** Department of Organizational Sciences and Communication, The George Washington University, Washington, DC, United States of America

* tbehrend@purdue.edu

**Data Availability Statement:** We are unable to provide the raw dataset for public use. In both the IRB and the informed consent documents provided to participants, we specifically noted that only

## Abstract

Educational virtual environments (EVEs) are defined by their features of immersion (degree of sensory engagement) and fidelity (degree of realism). Increasingly, EVEs are being used for career development and training purposes, which we refer to as career-oriented EVEs. However, little research has examined the effects of immersion and fidelity on career-related outcomes, like self-efficacy and interests, and the learning dynamics that may influence these outcomes. We address these research needs across two studies using an inductive approach. Study 1 compares welding career exploration in EVEs to traditional career exploration and finds that individuals using EVEs report more positive career self-efficacy. Study 2 examines the influence of social learning dynamics, or how individuals learn from each other through behavioral modeling, on performance and career-related self-efficacy and interest. Groups were assigned to use either a high or low immersion and fidelity EVE. Findings indicate strong social learning dynamics in both EVEs, but the effects were stronger for groups using the higher immersion and fidelity EVE. Specifically, groups converged on two performance measures, and the performance of individuals who were situated as behavioral models significantly predicted the performance of other group members. Performance at the individual level, in turn, predicted career self-efficacy and interest for men but not women, and only for those using the higher immersion and fidelity EVE. Based on these findings, we conclude with practical recommendations for and implications of implementing career-oriented EVEs for career exploration and skills training.

## Introduction

Educators are increasingly using immersive virtual reality (IVR) for a variety of education and career-related purposes. Often referred to as educational virtual environments (EVEs), these tools are employed in higher education, career exploration, and job skills training [1, 2]. EVEs

members of the research team would have access to the raw data. We did not stipulate an exception for the sharing of de-identified data. However, we are able to provide aggregated data for all variables in both studies including means, standard deviations, and correlations. All further requests for data should be made via the Office of Human Research at The George Washington University at 202-994-2715 (IRB protocol # 180402). All aggregated data and supplementary materials can be found at https://osf.io/32rgy/.

**Funding:** The author(s) received no specific funding for this work.

**Competing interests:** The authors have declared that no competing interests exist.

are especially advantageous in these capacities for fields like welding and aviation for which there are unique learning barriers. For example, it is challenging for educators to provide trainees with realistic job previews and opportunities to train alongside others due to the inherent risk of injury in these trades (e.g., burns, crashes), especially for novices. Career-oriented EVEs overcome these barriers and provide opportunities for career exploration and authentic job previews in a virtual, low-risk learning context. They also create safe settings for group-based skills training where trainees can engage in social learning by observing others and providing feedback [3].

EVEs have characteristics that make them distinct from other educational and training technologies; however, much remains unclear about how these features contribute to learning. Among the features of EVEs are their immersive nature and fidelity to real-life scenarios. Importantly, virtual environments can vary in their level of immersion and fidelity [4, 5]. In accordance with extant literature, we refer to EVEs with full immersion and high fidelity as virtual reality (VR) and those with relatively lower immersion and fidelity as desktop simulations.

Researchers have established links between higher immersion and fidelity and outcomes like engagement, enjoyment [6, 7], and training performance [8]. Relatively little research, however, has explored the associations between immersion and fidelity and important career-related outcomes like career self-efficacy and career interest. Similarly, few studies have examined the learning dynamics that may contribute positively to attitudes and performance in the contexts of EVE-based career exploration and skills training. As EVEs are increasingly used for these purposes, it is important to explore both the outcomes associated with EVEs in these contexts and the learning dynamics that explain these relationships.

We present two studies which each addressed one of these two research needs. In the first study, we addressed the need to explore the associations between immersion and fidelity and career-related outcomes by examining whether individuals report more positive career-related self-efficacy and interest when using VR for welding career exploration compared to individuals using desktop simulation or more traditional media (e.g., video, literature). The second study addressed the need to understand the learning dynamics that explain the affordances of EVEs. Specifically, we used groups engaged in welding skills training with either VR or desktop simulation to examine social learning dynamics and their influence on performance and career-related attitudes in this context. We then discuss the implications of our findings for the use of EVEs to aid in career development and skills training among adult learners.

We also note here that our research approach in both studies was inductive. That is, we posed exploratory questions to identify meaningful empirical relationships. This approach is useful for theory development and to address practical issues, particularly in fields characterized by rapid technological development [9]. For example, exploratory research has aided theoretical development and provided practical recommendations in domains like remote test proctoring [10]. Thus, IVR can benefit from inductive research because, as noted by Makransky and Petersen [11], there is still little theory to guide work in this field.

## Educational virtual environments

### Immersion and fidelity

EVEs are distinct from other training technologies in notable ways, namely immersion and fidelity [5]. *Immersion* in a virtual environment provides users with the feeling that they are participating in a comprehensive experience through the engagement of multiple senses [12]. In other words, an EVE is more immersive when it engages a higher number of senses or engages senses more intensely. EVEs used in practice vary in their levels of immersion. For example, VR that creates a 3-D, 360-degree environment through a head-mounted display is

more immersive than a desktop simulation that engages users through a 2-D computer screen. *Fidelity* is the degree of realism of a virtual environment. It provides users with the feeling that they are participating in a realistic experience [5]. Virtual environments vary in their level of fidelity; an EVE-based training has greater fidelity the more it accurately recreates the real-life environment.

## Attitudinal outcomes associated with immersion and fidelity

Research suggests that the experience of immersion in a virtual environment positively affects the attitudes of learners. Compared to traditional methods of instruction, individuals using highly immersive EVEs for training in a variety of STEM fields have demonstrated more positive attitudes and reactions such as engagement, interest, and motivation [7, 13]. Learners also have reported higher levels of engagement with material after learning through a desktop virtual environment [14]. Lastly, highly immersive EVEs are associated with increased self-efficacy in learners [6]. For example, Makransky and colleagues found more positive changes in self-reported intrinsic motivation, enjoyment, and self-efficacy related to performing laboratory safety protocols for students learning about laboratory safety in VR compared to reading about it.

While less research has focused on the effects of fidelity on learning and training outcomes, research shows that simulation fidelity is positively related to training transfer [15, 16] and may increase user acceptance of simulation-based education [17]. Some research also indicates that, like immersion, greater fidelity is associated with greater user engagement [18]. Overall, though, understanding of learner attitudes associated with the fidelity of EVEs is limited.

The extant literature generally supports the idea that immersion and fidelity lead to positive learner attitudes. Most existing studies, however, only compared EVEs to traditional forms of instruction (i.e., presence vs absence of immersion and fidelity). Thus, the effects of *varying* levels of immersion and fidelity is still largely underexplored [for exceptions see 19–21]. Moreover, although career-related attitudes are relevant outcomes to some of the ways that EVEs are being used in practice, only a select few studies have investigated such outcomes and have reported mixed results. One study found that EVEs improve career-related outcomes (e.g., outcome expectations) compared to traditional instruction [22], but another found no difference [23]. Further, both studies only compared EVEs to traditional learning. Thus, the effects that relative levels of immersion and fidelity have on career-related outcomes remain unexplored.

## Career exploration in VR and career-related attitudes

Though empirical evidence is largely absent or inconclusive, career exploration is an application of EVEs that shows potential in practice [24]. Adult learners and trainees use career-oriented EVEs to explore careers and develop skill in a variety of fields like surgery, aviation, and welding [8, 25, 26]. As described, EVEs are often used in such fields for practical reasons like safety.

Importantly, these virtual environments may also lower psychological barriers to exploring such careers by providing a low-stakes, low stress learning environment at a novice level [27, 28]. This notion is supported by research findings that learning in EVEs is associated with exploratory learning behaviors. Studies have shown that individuals demonstrate a willingness to take risks and make mistakes while learning in virtual settings more so than in traditional learning settings [13, 29]. By providing a low-risk exploratory learning environment, career-oriented EVEs give individuals opportunities to practice skills and develop positive attitudes toward careers that they may have otherwise not pursued.

Although the use of EVEs for career exploration shows promise for promoting positive career-related attitudes, research is needed that directly investigates this notion. Based on existing career exploration and career decision-making literature, the attitudes of career self-efficacy and interests are particularly important outcomes [30, 31]. Thus, we pose the following research question:

Research Question 1: Do individuals using EVEs for career exploration differ in career self-efficacy and interest from those using traditional methods for career exploration?

As noted, immersion and fidelity are defining features of EVEs that are associated with positive outcomes. Here, we explore whether using VR for career exploration is associated with higher career-related attitudes than using desktop simulation for career exploration, allowing for a direct comparison of EVEs that vary in immersion and fidelity. Therefore, we pose this second research question:

Research Question 2: Do individuals using VR for career exploration differ in career self-efficacy and interest from those using desktop simulation?

## Social learning in VR

Exploring the potential association between EVEs and career-related outcomes is an important first step. If we identify an association between immersion and fidelity and career-related outcomes, it then becomes necessary to identify the psychological mechanisms that explain this phenomenon. Identifying some of these mechanisms is the goal of Study 2. Recent scholarship addressing the question of how EVEs promote positive learning outcomes theorizes that instructional methods and context interact with technological features (e.g., immersion and fidelity) to facilitate the unique learning affordances of IVR, which then promote positive outcomes [11]. Because EVEs facilitate opportunities for skills training to take place in group contexts, social learning is relevant for understanding learning in EVEs. Broadly, social learning theory contends that observing the experiences of others can vicariously influence an individual's beliefs and their subsequent behavior [32]. Below, we review the research on technology-facilitated social learning. Then, as the basis for our third research question, we offer two competing propositions, grounded in current theory regarding learning in IVR [11], for how social learning dynamics may operate in groups training with EVEs. Lastly, we develop research questions to explore the relationships between performance and career-related attitudinal outcomes in this context.

### Social learning and performance

Existing research supports the idea that group-based learning experiences may influence outcomes positively (e.g., learning transfer) at the individual level [33]. Despite general support for the performance benefits of social learning, meta-analyses have identified several boundary conditions indicating that the social learning-performance relationship is complex. For example, the nature of the group learning experience moderates the relationship between group learning and academic performance. Specifically, in terms of academic performance, students in informal learning settings (e.g., students meeting on their own outside the classroom) benefited to a greater extent from social learning than students in structured, formal learning settings [34]. Another meta-analysis exploring the effects of social context on achievement in technology-mediated learning showed that the overall positive effect on individual outcomes varied based on the characteristics of the technology [35]. Differences in the amount of feedback provided and the extent of learner-control afforded by the training technology both influenced the relationship strength between group learning and performance. Overall, the

technological context in which learning takes place determines, in part, the degree to which social learning occurs and learners experience positive outcomes as a result.

## Immersion, fidelity, and social learning dynamics

If the degree to which social dynamics facilitate learning depends on the form of technology used, as evidenced in the research presented above, then studying how these dynamics operate when EVEs are used to facilitate group-based learning is imperative. In accordance with this existing scholarship, recent theory development in IVR posits that instructional methods and context interact with the technological features of IVR, such as immersion and fidelity, to influence learning outcomes [11]. The instructional context will promote positive outcomes to the extent that it facilitates the unique affordances of IVR, one of which is presence.

Based on this theoretical framework, we suggest that there are two competing possibilities for how social learning dynamics will operate in group-based training using EVEs. First, the feelings of presence that are facilitated by high immersion and fidelity may minimize social learning. The design of group skills training in EVEs often takes the form of trainees using the EVE one-by-one while others observe from the physical environment. The experience of being completely enveloped in a virtual environment while learning a skill could reduce trainees' awareness of others in the physical environment around them. Social learning would be diminished in this case and should not have much influence in facilitating the learning outcomes of EVEs. Alternatively, high immersion and fidelity may create an experience that sustains a high level of engagement among all group members even when they are not directly interacting with the EVE (i.e., in the physical environment). Social learning would be enhanced in this case, as each group member would act as a salient behavioral model for the rest of the group. These social dynamics, in turn, may facilitate the learning affordances of EVEs by allowing individuals to model their own learning experience based on the vicarious experience gained by observing authentic behavioral models (i.e., peers). If social learning dynamics are amplified, as in the second scenario, learning behaviors are likely to cluster within groups and there should be evidence of behavioral modeling. Conversely, a lack of group clustering and behavioral modeling would suggest little social learning. We pose the following research question to investigate these competing possibilities:

Research Question 3: To what degree does social learning take place during group-based skills training with VR and desktop simulation?

## Social learning, self-efficacy, and interest

If social learning does in fact influence learning behavior in group-based EVE skills training, then these dynamics should influence individuals' performance and, in turn, self-efficacy beliefs through vicarious experience [32]. That is, observing a similarly situated individual perform well or poorly will influence one's beliefs about his or her own likelihood of success or failure [36, 37]. This similarly situated individual is often referred to as a behavioral model for the task at hand. Schunk [38, 39] and Bandura and Schunk [40] showed that when learners observed a behavioral model successfully complete a cognitive task, their own feelings of efficacy toward that task improved.

Given the relationships between social learning, self-efficacy, and performance, one would expect that in learning environments that facilitate behavioral modeling, performance and self-efficacy should tend to cluster. That is, the success or failure of a behavioral model should influence the learning approach of observers through vicarious experience, which will then affect observers' performance and subsequent self-efficacy beliefs. This is likely to be especially true when individuals are engaged in a novel task. For such tasks, the absence of prior mastery

experiences means they are likely unable to draw on past models of success or their own prior behaviors to regulate self-efficacy expectations.

Beyond self-efficacy, the effects of social learning on performance are likely to influence trainees' interest in the content that was learned and their intention to pursue related content. That is, experiencing success in performing a task is predictive of further interest in that task [30, 41]. For example, Harackiewicz and colleagues [41] demonstrated a reciprocal relationship between situational (i.e., short term) and enduring (i.e., long term) interest and academic performance in a longitudinal study with adult learners. Thus, to the degree that social learning influences individual performance in group-based EVE skills training, it may also shape individuals' interests as a result.

Just as the degree to which social learning occurs within the context of group-based EVE skills training is unknown, the more specific influence of social learning on performance and its subsequent effects on self-efficacy and interest are also yet to be explored. It is important to examine these effects because, as we have explained, the degree to which social learning occurs has the potential to directly influence learner performance behaviors, self-efficacy, and interest. Thus, we pose the following research question:

Research Question 4: How does behavioral modeling in group-based skills training in VR and desktop simulation affect performance and, in turn, self-efficacy and interest?

## Overview of studies

We present two studies that each take an inductive approach to better understand the effectiveness of career-oriented EVEs to facilitate career exploration and group-based skills training, respectively. In Study 1, individuals using EVEs and traditional methods for welding career exploration were compared on career-related attitudes, which addressed Research Question 1. Those using EVEs were assigned to use either VR or desktop simulation, which allowed us to address Research Question 2 by examining differences in career-related attitudes as a result of higher immersion and fidelity. Study 2 filled the second research gap identified in this paper. We first examined the extent that social learning occurred in groups using EVEs for welding skills training, which answered Research Question 3. Second, we analyzed how social learning dynamics among groups affect performance and attitudes, which answered Research Question 4. We employed the same EVEs as Study 1 which allowed us to examine differences in the effects of social learning between EVEs varying in immersion and fidelity.

## Study context

We focus on the context of welding in both studies. Welding represents a profession at the human-technology frontier, incorporates many STEM principles, and is a middle-skills occupation that is critical to the manufacturing sector [42]. Further, welding is an appropriate context for our studies because EVEs are particularly useful for facilitating welding career exploration and skills training due to the inherent risk of injury (e.g., burns) with traditional training methods. EVEs have the potential to remove barriers in education and skills training for welding and other technical fields [3]. Below, we describe the two welding EVEs used in both studies.

### Equipment

**Virtual reality (VRTEX 360).** The VRTEX 360 is a VR welding device that replicates the environment of real-life welding. It includes a head-mounted display, a replica welding gun and mounted welding plate that are synched with the virtual environment, and a monitor for observers that displays in real-time what the welder sees. The head-mounted display provides feedback cues, and the monitor provides summary feedback after each welding trial. The

feedback cues include three icons that are visible to trainees while they weld. Each icon corresponds to an aspect of the weld (e.g., gun angle, gun distance from plate, speed) and provides synchronous regulatory feedback (e.g., a visual cue becomes increasingly red when the gun is too far or close to the plate, and lights green when the gun is at the correct distance). Following each weld, summary feedback is provided including performance score for several welding facets (e.g., welding angle, welding distance, welding speed) and an average score across all facets. Unlike welding cues, performance scores are not visible to welders in real time. To view these scores, welders navigate to a separate screen on the monitor.

**Desktop simulation (VRTEX engage).**   The VRTEX Engage is a desktop EVE. The VRTEX Engage is identical to VRTEX 360 in terms of the welding visuals, sounds, feedback cues, and performance scores provided; however, instead of viewing the virtual environment through a head-mounted display, trainees observe their weld on a computer screen. Thus, it is a less immersive experience and has less fidelity to the real-life welding environment and task. Trainees use a replica welding gun to weld across a plate that is just below the computer monitor that displays the weld in real time. As with the VR experience, following a weld, trainees can navigate to a feedback screen displaying performance scores.

## Study 1

### Participants and procedure

Study 1 was conducted with 119 adult undergraduates from a private east-coast university. The mean age of participants was 19.3 years old, 81% were female, and 49% were white. All participants were randomly assigned to one of five welding education conditions: literature, video, desktop simulation, short-exposure VR, and long-exposure VR. Across all conditions, participants were given a very brief introduction to welding via a 2-minute welding video, and then given 5 minutes (10 minutes for the long-exposure condition) to engage with their welding material or activity.

In the literature condition ($n = 22$), participants were given a packet with several brief articles about welding and how to weld and were instructed to read through as much as they could in the allotted time. Participants in the video condition ($n = 22$) were shown a 5-minute video in addition to the introductory video. This video presented a specific type of welding and the technique for performing it. Participants assigned to the VRTEX Engage ($n = 23$) practiced welding on a desktop welding simulation that simulates the experience of welding. Welders use a replica welding gun to weld across a plate that is just below the monitor, which displays the weld in real time. Finally, participants in the short exposure ($n = 40$) and long-exposure ($n = 12$) VR conditions practiced welding on the VRTEX 360 –fully immersive VR. Both the VR and desktop simulation provided participants with identical feedback about their performance and a hands-on virtual welding experience. Because the VR has a slightly longer learning curve, we included both longer and shorter durations to ensure that participants were not adversely affected by the longer orientation period.

After completing their activity, participants filled out a 10-minute questionnaire with demographic questions and the measures indicated below.

This study received IRB approval form from the Office of Human Research at The George Washington University. All participants read and signed informed consent documents before participating in the study.

### Measures

**Welding self-efficacy.**   A four-item self-efficacy scale was written for this study in order to measure welding career self-efficacy. Items for the scale were operationalizations of general

self-efficacy applied to the context of welding. Two example items are: "In regard to the job of welder: I have confidence in my ability to do the job" and "In regard to the job of welder: There are skills required of the job that I could not develop" (reverse coded). Each item was rated on a scale between *Strongly Disagree* (1) and *Strongly Agree* (5). The reliability of the measure was $\alpha$ = .62 (see online supplement for a factor analysis of this scale).

**Welding interest.** We employed a four-item self-report scale of welding interest. As with welding self-efficacy, the items were written specifically for the purposes of this study in order to measure welding interest. An example item is: "I would be interested in learning more about welding." Items were rated on a scale between *Strongly Disagree* (1) and *Strongly Agree*. The reliability of the measure was $\alpha$ = .82.

## Results

**Career attitudes.** Demographics for Study 1 are reported in Table 1 and descriptive statistics are reported for Study 1 outcome variables in Table 2. We began by addressing Research Question 1, which asked whether individuals using EVEs for career exploration differ in career self-efficacy and interest from those using traditional methods for career exploration. We conducted two ANOVA tests with welding condition as a between-subjects factor to test for differences in career-related attitudes (one participant in the literature condition and one participant in the desktop simulation condition failed to fully complete the self-efficacy items so that analysis was run with *n* = 117). Omnibus effects revealed significant differences between the five conditions for welding self-efficacy ($F(4, 112)$ = 2.83, *p* = .028, $h_p^2$ = .09), but not for welding interest ($F(4, 113)$ = .581, *p* = .677, $h_p^2$ = .02) (see Fig 1). Due to the insignificant omnibus test for interest, we only ran subsequent analyses for self-efficacy.

**Table 1. Demographic information and welding experience of student samples in Study 1 and Study 2.**

|  | Study 1 (*N* = 119) | Study 2 (*N* = 181) |
|---|---|---|
| **Gender** |  |  |
| Male | 19% | 30% |
| Female | 80% | 69% |
| Other | < 1% | < 1% |
| **Race/Ethnicity** |  |  |
| White, not of Hispanic origin | 51% | 58% |
| Hispanic | 8% | 12% |
| African American or Black | 8% | 9% |
| American Indian | <1% | < 1% |
| Asian or Pacific Islander | 33% | 19% |
| **Age** | *M* = 19.4, *SD* = 2.7 | *M* = 19.7, *SD* = 1.6 |
| 18–20 | 83% | 75% |
| 21–23 | 14% | 19% |
| 24–26 | < 1% | 4% |
| 27 and older | < 1% | 1% |
| **Previous welding experience** | *M* = 1.2, *SD* = .67 | *M* = 1.2, *SD* = .71 |
| Very inexperienced | 90% | 85% |
| Somewhat inexperienced | 5% | 7% |
| Neither inexperienced nor experienced | 2% | 4% |
| Somewhat experienced | 3% | 2% |
| Very experienced | < 1% | 1% |

*Note*. *M* = mean, *SD* = standard deviation.

**Table 2. Means, standard deviations, and correlations among Study 1 outcome variables.**

| Variables | M | SD | 1 |
|---|---|---|---|
| 1. Welding Self-Efficacy | 2.95 | 1.06 | – |
| 2. Welding Interest | 2.85 | 1.25 | .39** |

*Note. N* = 117. *M* and *SD* represent mean and standard deviation. *indicates *p* < .05.

**indicates *p* < .01.

Next, we conducted a linear contrast analysis to test for differences in self-efficacy between the EVE conditions (desktop simulation, short-, and long-exposure VR) and the literature and video conditions. This analysis directly addressed our first research question by comparing a linear composite of the sets of EVE conditions to a linear composite of the video and literature conditions. Results indicated a significant contrast whereby those in the set of EVE conditions reported higher self-efficacy ($t(112) = -2.65$, $p = .009$).

Lastly, we answered Research Question 2, which asked whether individuals using VR for career exploration differ in career self-efficacy and interest from those using desktop simulation. We conducted another linear contrast analysis between the desktop simulation condition and a linear combination of the short- and long-exposure VR conditions. Again, we only ran this analysis for self-efficacy due to the nonsignificant omnibus test for interests. Results did not show a significant difference in self-efficacy between the desktop simulation condition and VR conditions ($t(112) = -.58$, $p = .582$).

We also note here that an exploratory factor analysis of the 4-item self-efficacy scale demonstrated evidence that the scale was not unidimensional. In keeping with the exploratory nature of our study, we conducted supplementary analyses to examine the effects of immersion and fidelity on each factor of the self-efficacy scale. These analyses are presented in an online supplement.

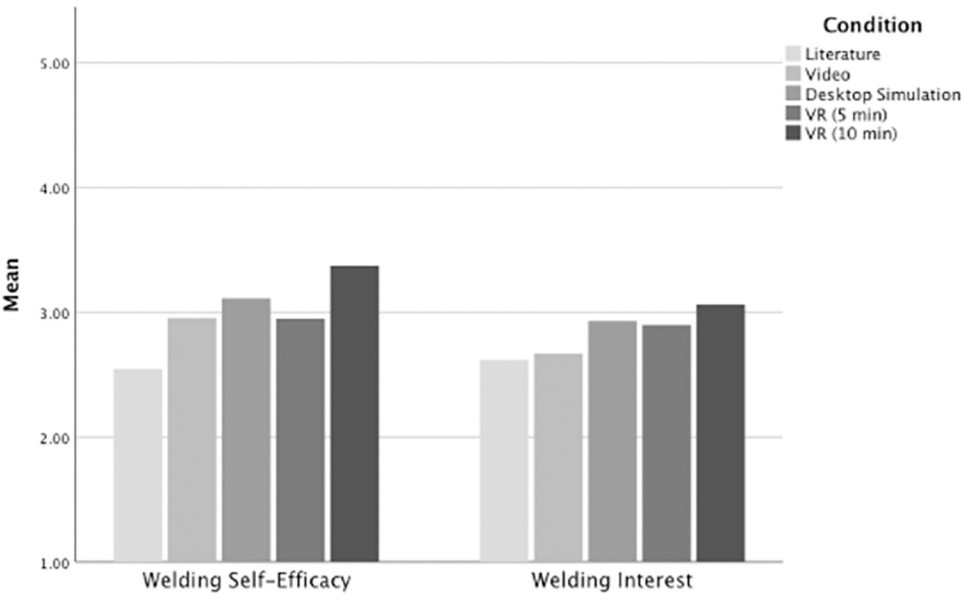

**Fig 1. Welding self-efficacy was higher for participants in the long-exposure VR condition.** Welding self-efficacy (left) and interest (right) by condition. The conditions are (from left to right) literature, video, desktop simulation, VR (5-min), VR (10-min).

## Study 1 discussion

Study 1 demonstrated that using EVEs for career exploration is associated with more positive welding career self-efficacy. Compared to reading or watching a video about the job of a welder, those who explored welding through EVEs reported significantly higher welding self-efficacy. No differences in attitudes emerged between those using VR and desktop simulation, suggesting no incremental benefit of the former compared to the latter. Overall, the pattern of findings suggests that using EVEs for career exploration is associated with greater self-efficacy regardless of the relative levels of immersion and fidelity.

## Study 2

The results of Study 1 led to the development of Study 2, which examined the use of career-oriented EVEs in the instructional context of group-based welding skills training. We explored the extent to which social learning occurs in this context (Research Question 3) and the role that it plays in influencing career-related outcomes (Research Question 4).

## Participants

Data collection took place in 2019 and 2020 at a mid-sized private university. Participants ($N$ = 181) were between the ages of 18 and 28, with a mean age of 19.7 years old. Participants received course credit for participation. The sample was 69% female, and 58% White, 19% Asian or Pacific Islander, 9% African American or Black, and 12% Hispanic. Participants reported their experience with VR and their experience with welding on a 5-point Likert scale, with 1 indicating a very little experience and 5 indicating a very high degree of experience. On average participants reported having very little experience with EVEs ($M$ = 1.60, $SD$ = .91) and very little previous experience with welding ($M$ = 1.24, $SD$ = .71), indicating that virtual reality welding largely represented a novel task for participants.

## Procedure

The VR and desktop simulation used in this study were the same as Study 1. Participants were assigned to groups of three or four when they registered for the experiment. Groups were assigned to use either the VR or desktop simulation. Groups began by watching a brief introductory video on welding. Following this video, the facilitator provided an orientation for proper use of the hardware and navigation of the software on the VR or desktop simulation. A full welding demonstration was not given to ensure that group members served as behavioral models for each other. A single weld typically takes 30 seconds to one minute. Though there was variability in the time it took to complete one weld, five minutes was sufficient for participants to complete several welds ($M$ = 4.18, $SD$ = 0.18). Each group member, one at a time, was provided a five-minute interval to weld. As one member in a group welded, all other members were positioned around the welder to be able to observe their behaviors and performance (mimicking a real group training environment). In both the VR and the desktop simulation conditions, non-welding group members were able to simultaneously observe the welding member directly (e.g., their body positioning and hand movements) while also viewing the welding member's virtual/simulated welds on a highly visible monitor. Additionally, all participants were free to encourage and give feedback to each other throughout the experiment. During their five minutes of welding, participants were allowed to ask for feedback and to check the feedback screen as often as they wanted. To promote a learning motivation among these participants who were not real-life welding skills trainees, facilitators told participants to try to improve their skills on each weld, and that higher scores indicated better performance.

Data were collected from two sources during the study. First, objective performance scores for every welding trial were stored. Second, after each participant completed their 5-minutes of welding they were instructed to complete a 10-minute questionnaire including demographic questions, attitudinal measures (same as in Study 1), and open-ended questions (see online supplement).

This study received IRB approval form from the Office of Human Research at The George Washington University. All participants read and signed informed consent documents before participating in the study.

## Measures

**Welding performance.** Each welding trial was objectively scored on several technical welding criteria (e.g., welding angle), and these welding criteria were averaged to create an overall performance score for each trial. As mentioned, all performance scoring was automated. Performance was scored from 0 to 100 with 0 indicating the poorest performance and 100 indicating the best performance. Participants' final performance scores were calculated as the mean of their overall performance scores across all of their welding trials.

**Learning strategy.** Learning strategy is another indicator of trainees' learning behavior while using the EVE. It was operationalized as the number of trials that a participant completed in the 5-minute time period. This continuous variable was conceptualized as indicating whether an individual took a more methodical approach or an approach of trying to complete a high-volume of welds. Fewer trials indicated that an individual proceeded slowly, likely stopping between each trial to monitor their feedback and ask questions of their peers or the experimenter. A higher number of trials indicates that a participant attempted to fit in as many trials as they could during the 5-minute period, hoping to improve through a greater volume of practice.

**Welding self-efficacy.** Participants in Study 2 completed one of two welding self-efficacy scales depending on whether they were assigned to VR or desktop simulation. We revised the self-efficacy scale between the collection of data for the VR and desktop simulation conditions to improve its psychometric qualities and the internal validity of this part of the study (see the online supplement for more detail). Since these scales differed, any direct comparison between VR and desktop simulation for self-efficacy in Study 2 is inappropriate. Individuals assigned to VR ($n$ = 131) completed two items measuring welding self-efficacy that were taken from the four-item scale used in Study 1. These items asked participants about their confidence in developing welding skill, which was of primary interest in this study, whereas the other pair asked about participants' current confidence in their ability to do the job (e.g., "I have confidence in by ability to do the job") (see the online supplement for the factor analysis of the full 4-item scale). The two scale items are: "In regard to the job of welder: I could develop all the skills needed to perform the job well" and "In regard to the job of welder: There are skills required of the job that I could not develop" (reverse coded). Participants rated their agreement with each statement from *Strongly Agree* (1) to *Strongly Disagree* (5). The correlation between these items was $r$ = .56.

Participants assigned to desktop simulation ($n$ = 50) completed a seven-item welding self-efficacy scale. This measure was adapted to the context of welding from a scale of self-efficacy for a college biology course that has been validated in previous work [43]. Participants rated their confidence in performing the task described in each item from *Not Confident* (1) to *Extremely Confident* (5). Example items are: "Understand the material taught in a welding course", "Develop the skills of a professional welder", and "Comprehend the scientific and theoretical aspects of welding". The reliability of this scale is α = .91.

**Welding interest.**   The interest scale used in Study 2 was the same scale used in Study 1 (see Study 1 Measures section).

## Analysis and results

Data from several participants who did not complete measures or chose not to participate in the welding simulation were removed before analysis. Of the final sample ($N = 181$), $n = 131$ ($K = 42$) used the VR and $n = 50$ ($K = 14$) used the desktop simulation. Unbalanced data were due to a disruption in data collection in the spring of 2020 due to the COVID-19 pandemic. We performed a series of exploratory quantitative analyses to address our third and fourth research questions. Additionally, a supplementary qualitative analysis of participants' open-ended comments was conducted to provide a more detailed account of their attitudes toward welding as career and toward learning with virtual environments (see online supplement).

Demographics and descriptive statistics for each variable are reported in Tables 1 and 3, respectively. Of note, average participant welding performance significantly differed between the two conditions. Those using the desktop simulation ($M_{performance} = 72.92$, $SD = 14.91$) performed better ($t(180) = 6.44$, $p < .001$) than those using VR ($M_{performance} = 49.19$, $SD = 15.70$), indicating that the highly immersive environment may add an extra layer of difficulty for those inexperienced in both welding and virtual reality. Participants' open-ended comments support this notion (see online supplement). Given the study's exploratory nature, sample size differences, and difficulty differences, we analyzed the VR and desktop simulation groups separately rather than including condition as a moderator.

**Group effects.**   We addressed Research Question 3, which asked to what degree social learning takes place during group-based skills training with VR and desktop simulation, by examining the degree to which our objective measure of welding performance and learning strategy clustered by group membership. We calculated the intraclass correlation coefficient (ICC (1)) for both criteria. These variables did not significantly relate to one another, suggesting that they should be analyzed separately. An ICC reports the extent to which variables for individuals within a group correlate with or resemble each other, indicating the strength of the group effect for an outcome. For participants who used the VR, learning strategy (ICC = .61) and welding performance (ICC = .48) displayed strong group effects. Differences between

**Table 3.  Means, standard deviations, and correlations among Study 2 variables.**

| Variable | M | SD | 1 | 2 | 3 | 4 | 5 | 6 | 7 | 8 | 9 |
|---|---|---|---|---|---|---|---|---|---|---|---|
| 1. Condition | 1.27 | 0.45 | – | | | | | | | | |
| 2. Gender | 0.69 | 0.46 | -.01 | – | | | | | | | |
| 3. VR Experience | 1.63 | 0.93 | .06 | .00 | – | | | | | | |
| 4. Welding Experience | 1.24 | 0.70 | .13 | .02 | .18* | – | | | | | |
| 5. Group Size | 3.37 | 0.63 | .20** | .00 | .04 | .04 | – | | | | |
| 6. Learning Strategy (Total Welding Trials) | 7.48 | 3.91 | .15* | -.08 | -.09 | .05 | .07 | – | | | |
| 7. Welding Performance | 55.65 | 18.73 | .57** | -.14 | .15* | .23** | .23** | .07 | – | | |
| 8. Welding Self-Efficacy (2-item) | 3.71 | 0.96 | NA | -.30** | .03 | .13 | .01 | -.12 | .22* | – | |
| 9. Welding Self-Efficacy (7-item) | 2.27 | 0.94 | NA | -.42** | .13 | .26 | .00 | -.27 | .20 | NA | – |
| 10. Welding Interest | 2.81 | 1.11 | .06 | -.26** | .17* | .27** | .01 | .05 | .18* | .39** | .55** |

*Note*. $N = 181$. For welding self-efficacy (2-item), $n = 131$. For welding self-efficacy (7-item), $n = 50$. $M$ and $SD$ represent mean and standard deviation. Condition: 1 = VR, 2 = desktop simulation. Gender: 0 = male, 1 = female.

* indicates $p < .05$.

** indicates $p < .01$.

teams accounted for 61% of variance in strategy and 48% of variance in performance. ICCs were relatively lower for the desktop simulation condition. Differences between groups in strategy (ICC = .33) and performance (ICC = .26) accounted for 33% and 26% of variance, respectively. Within group variance in strategy and performance were similar in both conditions but between group variance was much greater for groups who used the VR. Overall, group clustering of welding performance and learning strategy indicated that social learning was present to a large extent in the VR condition, and to a lesser extent in the desktop simulation condition.

**Behavioral modeling.** We took several steps to explore Research Question 4, which asked whether behavioral modeling in group-based skills training in VR and desktop simulation affects performance and, in turn, self-efficacy and interest. First, we explored the effects of behavioral modeling within the welding groups using a linear mixed effects model. The first welder (i.e., lead welder) was considered a salient behavioral model for all group members because this individual was the first example of welding performance and learning strategy. Therefore, the strategy and performance of each group member was regressed on the lead welder's strategy and performance, respectively. Group size and gender were initially examined as control variables, but both were not significant predictors and were removed. Group membership was included as a higher order variable in each model.

The mixed effects model indicated that for VR ($n$ = 89, $K$ = 41), the strategy ($b$ = .57 [95% CI: .32; .81], $p$ < .001) and performance ($b$ = .46 [95% CI: .22; .71], $p$ < .001) of the lead welder significantly predicted subsequent group member strategy and performance. For the desktop simulation groups ($n$ = 35, $K$ = 14), lead welder strategy predicted subsequent group member strategy ($b$ = .49 [95% CI: .18; .80], $p$ = .003); however, lead welder performance did not significantly predict subsequent group member performance ($b$ = .30 [95% CI: -.11; .70], $p$ = .160). These results suggest, for the most part, that the learning strategy and the welding performance of the lead welder set the course for group members that followed.

Next, we examined the degree that the learning strategy and welding performance of the lead welder influenced participants with multiple behavioral models (i.e., two or more group members welded prior to their turn), accounting for the strategy and performance of other behavioral models. In other words, did the lead welder stand out as a behavioral model for those with the chance to observe multiple behavioral models? Linear mixed effects models were again used. Only participants with more than one behavioral model were analyzed and, due to the small sample, we only analyzed the VR condition. We regressed the learning strategy of group members with multiple behavioral models on the learning strategy of the lead welder and of the behavioral model that welded immediately before them, with group membership included as a higher order variable. The same model was used for welding performance.

Results indicated that, after controlling for the strategy and performance of the most immediate behavioral model, the strategy ($b$ = .13 [95% CI: -.11; .37], $p$ = .278) and performance ($b$ = .21 [95% CI: -.05; .46], $p$ = .110) of the lead welder was no longer predictive of the strategy and performance of subsequent group members ($n$ = 48, $K$ = 35). On the other hand, results indicated that the learning strategy ($b$ = .67 [95% CI: .43; .91], $p$ < .001) and welding performance ($b$ = .55 [95% CI: .29; .80], $p$ < .001) of the most immediate behavioral model were highly predictive. Thus, while the lead welder set the course for the group, the most salient behavioral model for individual participants tended to be the peer that welded immediately prior.

**Performance as a moderator.** We explored the possibility that the welding performance of behavioral models influenced the degree to which subsequent group members imitated their learning strategy. In other words, did the tendency to imitate the most immediate

behavioral model's learning strategy depend on how well the model performed? Using a linear mixed effects model, we regressed the learning strategy of each group member on the learning strategy of the group member that welded immediately before them, including an interaction term between the behavioral model's performance and learning strategy. No significant interaction was observed for either the VR ($b$ = -.18 [95% CI: -.38; .02], $p$ = .084) or desktop simulation ($b$ = -.34 [95% CI: -.72; .05], $p$ = .085). The behavioral model's average welding performance was not found to significantly moderate the relationship between their own learning strategy and subsequent group member learning strategy. These results suggest that group members tended to imitate the learning strategy of behavioral models regardless of whether they performed well or poorly. This finding was striking and is discussed further in a later section.

**Career-related attitudes.** To examine the self-efficacy and interest components of our fourth research question, we determined the direct effect of welding performance on self-efficacy and interest in welding. A linear mixed effects model was again used to account for the nested data. Self-report measures for welding self-efficacy and welding interest were both individually regressed on participant welding performance. We included gender as a control variable because gender was strongly related to performance, interest, and efficacy. Results showed that while controlling for gender, performance in the VR condition did not significantly predict welding efficacy ($b$ = .15 [95% CI: -.03; .23], $p$ = .095) or welding interest ($b$ = .09, [95% CI: -.09; .26], $p$ = .327), whereas performance on the desktop simulation did predict welding efficacy ($b$ = .38, [95% CI: .13; .63], $p$ = .004), but not interest ($b$ = .17 [95% CI: -.12; .46], $p$ = .254). To examine these findings in more detail, we included gender as a moderator in the relationships instead of as a control variable. Male was coded as 0 and female as 1. For the VR there was a significant interaction effect between gender and performance in predicting both welding efficacy ($b$ = -.23 [95% CI: -.40; -.06], $p$ = .004) and welding interest ($b$ = -.14, [95% CI: -.32; .03], $p$ = .001). The relationship between performance and welding self-efficacy and interest were stronger for males than females (See Fig 2). These same results were not observed for the desktop simulation condition (see Fig 3), where gender did not moderate the relationship between performance and welding self-efficacy ($b$ = .07 [95% CI: -.17; .32], $p$ = .340) or interest ($b$ = .009 [95% CI: -.26; .28], $p$ = .885).

## Study 2 discussion

The purpose of Study 2 was to investigate the degree to which social learning takes place in group-based skills training using VR and desktop simulation, and how social learning dynamics shape performance and attitudinal outcomes. We posed two research questions and addressed them with specific analyses driven by observation and previous research on social learning theory.

Regarding Research Question 3, results provide evidence for the presence of social learning dynamics in varying degrees for both the VR and desktop simulation conditions. In both conditions, groups converged on performance and learning strategy. Those in the VR condition, though, demonstrated significantly higher convergence on both metrics, indicating a higher degree of social learning. Second, we found that the welding performance and strategy of the group's lead welder strongly predicted the rest of the group members' welding performance and strategy, which demonstrates the effect of behavioral modeling. This finding prompted a more detailed investigation of effects of behavioral modeling. We next examined participants who observed multiple behavioral models (i.e., welded third or fourth) to see if they were more strongly influenced by the person who welded right before them than the first welder. Analyses showed that for these individuals, the welder who went right before them strongly predicted

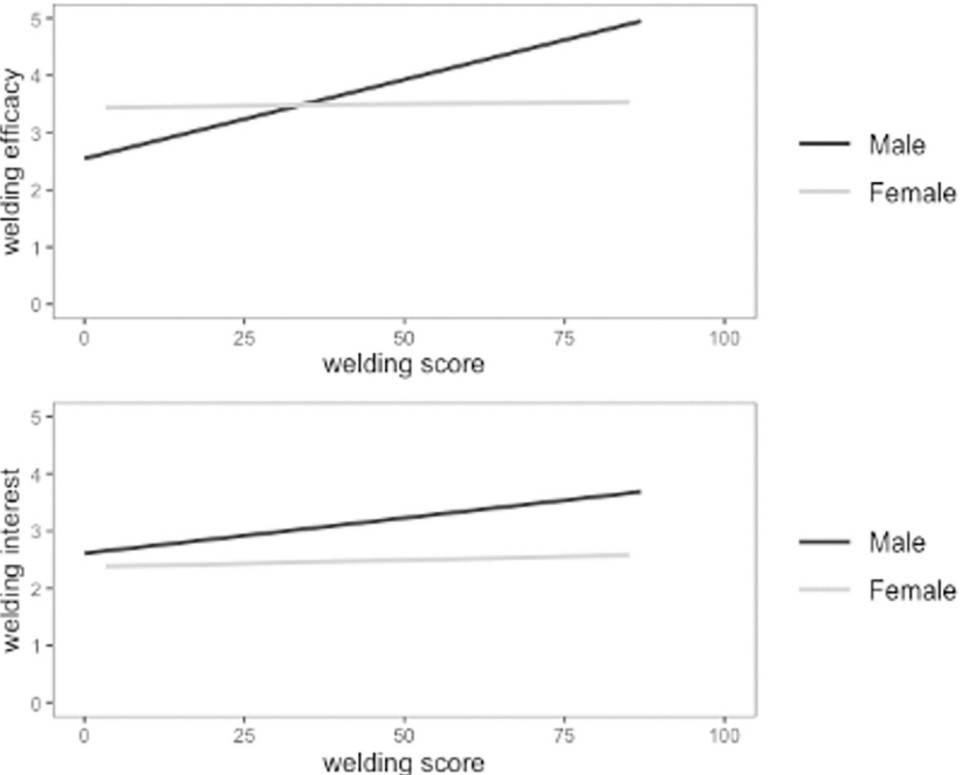

**Fig 2. Performance predicts welding self-efficacy and interest stronger for males than females in the VR condition.** Interaction effect of gender and performance on welding self-efficacy (top) and interest (bottom) for VR. The darker line represents males and the lighter line represents females.

their welding performance and learning strategy while the first welder did not. Taken together, these findings indicated that in both conditions the first welder set the tone for the group but that individuals were more directly influenced by the performance and strategy of the welder right before them.

The final aspect of behavioral modeling that we explored was whether an individual's level of welding performance affected whether subsequent group members modeled their learning strategy. Surprisingly, analyses revealed a nonsignificant interaction between a behavioral models' learning strategy and performance in predicting the learning strategy of subsequent group members. We interpreted these findings as a tendency for participants to model their own learning strategy after the most proximate behavioral model regardless of how well that model performed overall.

Next, we address the career-related attitudinal outcomes. Results demonstrated that welding performance was only predictive of male participants' welding self-efficacy and interest in the VR condition. These results indicate that performance in group-based EVE skills training, which is evidently influenced by social learning dynamics, in turn influences welding self-efficacy and interest for only a certain subset of trainees; namely, males using high immersion and fidelity VR.

## General discussion

In two separate inductive studies we explored the use of EVEs for the purposes of career exploration and group-based skills training. By creating a safe learning environment for a variety of

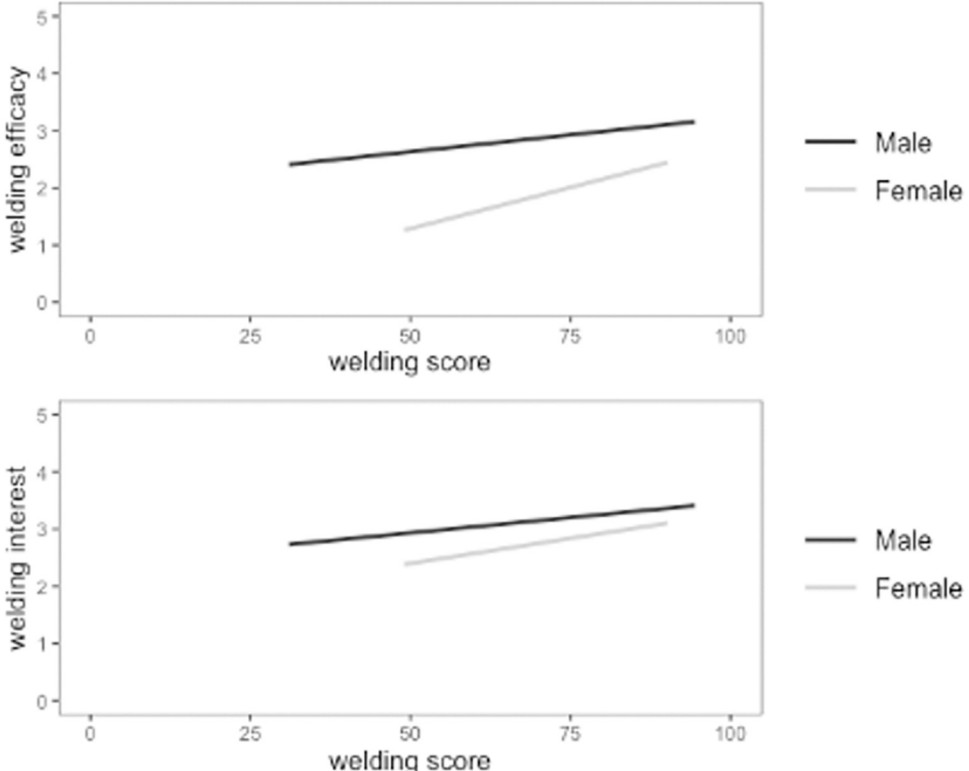

**Fig 3. Gender did not moderate the relationships between performance and welding self-efficacy and interest in the desktop simulation condition.** Interaction effect of gender and performance on welding self-efficacy (top) and interest (bottom) for desktop simulation. The darker line represents males and the lighter line represents females.

skill levels and removing barriers to group-based learning, career-oriented EVEs have the potential to greatly expand learning opportunities. The cumulative results of these studies demonstrate that EVEs may be useful training tools in these capacities, but they also highlight some potential pitfalls particularly in the context of group-based training.

For career exploration, providing adult learners with the opportunity to experience welding through an EVE was associated with greater expressions of self-efficacy than traditional methods of career exploration. This suggests that highly engaging and realistic career exploration experiences can bolster feelings of agency toward a particular job more so than simply reading or watching videos about that job. Welding interest was not significantly higher for those using VR and desktop simulation. The reason may be attributable to our sample; largely affluent university students may report low interest in welding regardless of how they experience it. If so, our findings may represent a lower bound of effects that would be larger in other samples. Lastly, neither career-related attitude differed between those using VR and desktop simulation. Thus, there may be a ceiling for the positive effects of immersion and fidelity for these outcomes. Career exploration using EVEs compared to traditional media appears to be beneficial, but the relatively minor increase in immersion and fidelity of VR compared to desktop simulation does not seem to provide an incremental benefit. Our findings highlight the potential utility of EVEs for promoting self-efficacy in technical careers and help to clarify the career-related outcomes associated with the IVR features of immersion and fidelity.

Based on Study 2, we conclude that social learning dynamics are strong in group-based EVE skills training. These findings are significant because, as we explained previously, it was

theoretically reasonable to argue that immersion and fidelity would have negative effects on social learning dynamics by distracting learners from the task at hand and from the observation of others, rather than increasing their engagement with both. The clustering of welding performance and strategy and the effects of behavioral modeling were remarkable in both conditions, but they were even stronger in the VR than the desktop condition, providing significant evidence that higher immersion and fidelity EVEs enhance social learning.

Study 2 also showed that participants tended to model their learning strategy based on peers that welded before them, even when those models performed poorly. Thus, whether this training experience improved performance was largely influenced by the level of proficiency demonstrated by the behavioral model(s) that trainees observed. This finding is in line with past research on social learning strategies. People do not engage in social learning indiscriminately, but rather tailor their strategies based on the learning context [44]. A salient characteristic of the learning context in our study was participants' unfamiliarity with welding and virtual environments. In uncertain contexts (i.e., little prior task information), trainees tend to rely more on social information when performing a task [45]. This idea has been supported by research with novice learners [46], showing that they tend to incorporate exemplars even when told that the example was ineffective or inappropriate. Our findings of strong social learning effects are in line with this existing literature. This tendency has clear implications for individual and group skill development, and subsequently career decision-making, which we discuss in the section on practical implications.

Importantly, uncertainty in the learning context does not explain the larger social learning effects observed for groups using VR compared to desktop simulation. In the development of our research questions, we presented two possibilities whereby the use of EVEs could plausibly enhance or diminish social learning. Our results suggest a positive relationship between social learning and immersion and fidelity. Additionally, these findings support propositions made by the CAMIL framework, which posits that instructional context interacts with the features of IVR to influence learning [11]. In the case of the present study, the IVR features of immersion and fidelity interact with the instructional context of behavioral modeling. Namely, in this context there is a positive relationship between the level of these features and social learning. In turn, social learning seems to predict career-related attitudes in specific cases (e.g., males using VR), which may point to potential boundary conditions for relationships predicted by the CAMIL framework. As we mentioned previously, the CAMIL framework also argues that the link between IVR features of immersion and fidelity and learning outcomes is the psychological construct of presence [11]. Although we did not examine presence as a mediator in the current study, future research should do so in order to test this proposition of the CAMIL framework.

Lastly, performance did have some predictive ability for self-efficacy and interest; however, the effects were limited to males using VR. This finding is in line with other research finding gender differences in student outcomes with EVEs [20] and is significant in the context of the current study because a gender gap already exists in technical fields with traditional training methods [47]. If performance in group-based skills training in EVEs only increases career self-efficacy for males, there is the potential for its use to widen the gender gap. We encourage future research to explore this possibility in greater depth.

Finally, we want to highlight the relative brevity of the learning experience that participants in our study had compared to what would be the norm in technical skills training or career and technical education (CTE) programs. Each participant had just 5-minutes of direct interaction with the VR or desktop simulation and then another 10–15 minutes of observing others for a total of 20 minutes of combined direct and vicarious learning. The fact that this brief experience had significant effects on performance, learning strategy, and (to a lesser extent)

attitudes speaks volumes about the potential of group training facilitated by career-oriented EVEs. We discuss this practical potential more in the next section.

## Practical implications

The first practical implication of our findings pertains to the implementation of EVEs to facilitate group-based training in CTE or other technical skills training. Our results show that if learners are presented with a novel task in such a context, they tend to follow the example of the task behavior that is modeled for them, regardless of its effectiveness. Consequently, it is imperative that instructors provide trainees with authentic and credible task behavior modeling. This model could be the instructor, a subject or task expert, or another trainee who has prior experience and expertise. The importance of expert behavioral modeling for training in EVEs to be successful is echoed by other scholars [12]. Our findings also indicate that instructional design for group-based skills training in VR should consider the ordering of participants to maximize the positive (and minimize the negative) effects of behavioral modeling. For example, trainees who are known to be worse performers (or who have less experience) should follow as closely as possible those who are known to be better performers. Importantly, due to our use of a general student sample, future research that generalizes our findings to groups of technical skills trainees using EVEs is needed. We discuss this idea more in following section.

A second implication of our study has to do with career decision-making and pipeline development for technical careers. Social cognitive career theory (SCCT) [30, 31] proposes that self-efficacy and career interests are two of the primary factors that lead to the setting of career goals and ultimately intentions to pursue those careers. Based on this research, particularly Study 1, exploring a technical job or learning technical skills in EVEs may be advantageous for career self-efficacy. As we discussed earlier, EVEs also lower psychological barriers to engaging in career exploration by creating a low-stakes, low stress learning environment for novices. Thus, EVEs present opportunities for students to develop self-efficacy for careers that they would not have otherwise considered. Further, SCCT is cyclical in nature, meaning that repeated positive experiences reinforce beliefs and attitudes. Overall, if implemented correctly with authentic behavioral modeling, EVEs show promise as a tool for facilitating individual career pursuit and, consequently, talent pipeline development for important technical jobs like welding.

## Limitations and future research

There are a few potential limitations to the findings of this study to consider. The first has to do with the generalization of these findings to other relevant populations. We drew our sample from a population of undergraduate psychology students rather than from a population of real-life welders or trainees. The purpose of the current study was to provide initial evidence of the affordances of EVEs for the outcomes of career exploration and skills training, and the effects of social learning dynamics in this context. Studies that establish phenomena empirically in a controlled setting are an important precursor to field research. Thus, a student sample was appropriate for our goals; however, our results may be limited in their applicability to people who are actually participating in welding or other CTE training programs. The primary differences between these populations are prior experience with welding or other technical disciplines and motivation. We encourage future research exploring the use of career-oriented EVEs to facilitate career exploration and group-based skills training in CTE programs.

Another limitation lies in our measurement of welding self-efficacy. Since no such prior measure existed, we were required to devise our own. While we attempted to write the items for this measure based off prior measures of contextual self-efficacy, the four-item measure

was not unidimensional. To further address this issue, we provided an online supplement that presents the factor analysis of this scale and several supplementary analyses using the separate dimensions of self-efficacy. A second caveat in regard to self-efficacy is that the directionality of the relationship between self-efficacy and performance is somewhat contentious [48, 49]. Some have resolved this contention by conceptualizing feedback loops. That is, the observation of others influences one's self-efficacy beliefs, which in turn influence task approach, and resulting performance, which then further influences self-efficacy. For instance, some have found evidence for *efficacy-performance spirals* [50]. According to these findings, self-efficacy and performance relate to each other over time such that initial self-efficacy influences performance, which influences subsequent self-efficacy accordingly, and so on. These ideas should be kept in mind when interpreting our findings regarding the relationship between performance and self-efficacy.

## Conclusion

In these two studies we sought to investigate the affordances of career-oriented EVEs for welding career exploration and group-based skills training, and the role that social learning plays in facilitating these learning affordances. We discovered that EVEs are a potentially useful tool for both purposes but that there are potential pitfalls that must be considered by educators and further studied by scholars. Our findings inform theory and practice regarding the use of IVR for education and training purposes and help to clarify the effectiveness of EVEs for promoting career-related outcomes. Further, our findings expand the literature on the learning outcomes associated with varying levels of immersion and fidelity as features of EVEs.

## Supporting information

**S1 File.**
(DOCX)

**S2 File.**
(DOCX)

## Acknowledgments

The authors wish to thank current and former members of the WAVE Lab for their help in developing this paper, especially Charlotte Shephard and Ian Siderits. We also thank Dr. Ryan Watkins for assisting with the logistics for this study and Dr. Jason Scales for making equipment and resources available for this study. Portions of these findings were presented at the 2019 APA Technology, Mind, and Society Conference in Washington, DC. Correspondence concerning this article should be addressed to: Tara S. Behrend, Department of Psychological Sciences, Purdue University, 703 Third Street, West Lafayette, IN 47907. Email: tbehrend@-purdue.edu.

## Author Contributions

**Conceptualization:** Daniel M. Ravid, Tara S. Behrend.

**Data curation:** Bradley D. Pitcher, Daniel M. Ravid.

**Formal analysis:** Bradley D. Pitcher, Daniel M. Ravid, Peter J. Mancarella.

**Methodology:** Bradley D. Pitcher, Tara S. Behrend.

**Project administration:** Daniel M. Ravid.

**Supervision:** Bradley D. Pitcher, Tara S. Behrend.

**Visualization:** Bradley D. Pitcher.

**Writing – original draft:** Bradley D. Pitcher, Daniel M. Ravid.

**Writing – review & editing:** Bradley D. Pitcher, Peter J. Mancarella, Tara S. Behrend.

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
