## [Decision Letter · Decision Letter 0]

5 Apr 2022

PONE-D-21-37956Social learning dynamics influence performance and career self-efficacy in career-oriented educational virtual environmentsPLOS ONE

Dear Dr. Pitcher,

Thank you for submitting your manuscript to PLOS ONE. After careful consideration, we feel that it has merit but does not fully meet PLOS ONE’s publication criteria as it currently stands. Therefore, we invite you to submit a revised version of the manuscript that addresses the points raised during the review process. Specifically, the reviewers expressed concerns about the focus of the study, clarification of the procedure as well as more critical discussions of the findings.

We look forward to receiving your revised manuscript.

Kind regards,

Mingming Zhou, Ph.D.

Academic Editor

PLOS ONE

Journal Requirements:

Reviewers' comments:

Reviewer's Responses to Questions

**Comments to the Author**

1. Is the manuscript technically sound, and do the data support the conclusions?

Reviewer #1: Partly

Reviewer #2: Yes

Reviewer #3: Partly

Reviewer #4: Yes

2. Has the statistical analysis been performed appropriately and rigorously? 

Reviewer #1: Yes

Reviewer #2: Yes

Reviewer #3: Yes

Reviewer #4: Yes

3. Have the authors made all data underlying the findings in their manuscript fully available?

Reviewer #1: Yes

Reviewer #2: Yes

Reviewer #3: Yes

Reviewer #4: No

4. Is the manuscript presented in an intelligible fashion and written in standard English?

Reviewer #1: No

Reviewer #2: Yes

Reviewer #3: Yes

Reviewer #4: Yes

5. Review Comments to the Author

Reviewer #1: The paper is very ambitious, particularly given the fact that it is based on two studies. However, the present version of the paper demands a high degree concentration and focus or a significant cognitive load. This is because it is very difficult to grasp its key ideas and objectives. Most specifically, it was a challenging task keeping in mind the focus on both affordances of career-oriented EVE and the role that social learning play in facilitating learning affordances of the studies --- while trying to digest four research questions at the same time.

Study 2 appears to be strongest and the one that's better developed. Therefore, I recommend placing a stronger focus on it. Then, perhaps study 1 can be repurposed/repackaged as a conference paper.

Other adjustments needed are:

1 – Revision of the papers' abstract. It needs to be edited for clarity and readability. The verbiage used in some of the sentences is a bit unorthodox. Language used should be familiar, precise and unambiguous.

2 – Inclusion of a demographics table. This will help the reader understand, at a glance, who the participants in the study(ies) are and gain a better perspective of their backgrounds. Some of that information is buried in the text. So, it’s hard to keep up with who is involved in the study.

3 – Break table 2 into small ones and place those where they are needed. Not sure all of the correlation figures are needed to be shown in one table. This would also make it possible to reduce or eliminate the amount of lookbacks to table 2 as the text is being read.

Reviewer #2: This was an interssting study concerning two variables as it realtes to Self Efficacy and the usage of AI orm Visual Reality. the Research Deisgn was sound...The Research Questions came directly from the Literature. The statistical analysis was sound although some instruments created were not reliable for valid.

Overall this was well done and impressive for a lead author who is un undergraduate student.

Reviewer #3: Review of “Social learning dynamics influence performance and career self-efficacy in career-oriented educational virtual environments”

PLOS ONE

MS# PONE-D-21-37956

Comments for the Author(s)

This paper addresses an interesting question of how virtual environments may enhance and influence the learning and career views of students. Using welding as a skill to be learned, the authors look at the impact of virtual reality on performance and career attitudes.

My major concern with your paper (which the authors discuss) is that the learning being studied is welding training, but the samples for both studies are undergraduate psychology students. I think that the characteristics of the sample may have little impact on styles of learning and performance, but question what college undergraduates have to say about self-efficacy in a welding career. A sample from a community college or vocational school would have been much more useful. The sample may have also contributed to the testing issues with the career self-efficacy scale.

Another concern is that most readers would expect to find that as EVEs become more immersive (more virtual) they should have more positive effects. In the same way that a video will be better than a picture, we would expect virtual reality to be more effective than a desktop simulation, which should be more effective than watching a video. Can you make a stronger case for why your findings are important? For example, can your findings say anything about the CAMIL model you cite (Makransky & Peterson, 2021)?

More Specific Issues:

1. I thought that the beginning of your introduction (pages 3-4) had a very good justification for studying EVEs.

2. You emphasize in the introduction that you are presenting inductive studies with exploratory research questions, Is this why you are not submitting it to a psychology journal? It is not theoretical enough?

3. I thought that your discussion of immersion and fidelity on page 5 was overly theoretical and so not as clear as it could be. Is immersion focused on seeing and fidelity focused on touching?

4. You mention that previous research has examined self-efficacy (Makransky et. al., 2019) but don’t describe this research. I assume it examined self-efficacy of the task, not career self-efficacy, but some discussion of it seems warranted.

5. I wasn’t sure if the group of learners were together IN the EVE when you discussed the possible ways learners might model each other’s behaviors (pages 9-10). This was compounded in your methods section (page 18) where it wasn’t clear to me how the rest of the group observed the learner who was practicing. You need to make this procedure clearer. I think what happened is that each subject took a turn with the virtual goggles (or the monitor) to work on the welding simulation. Were the others physically with them while they practiced or did they watch through a monitor? After each subject practiced welding, did they observe the others, or did they fill out the questionnaire?

6. When I first read your discussion on Page 11, it wasn’t clear to me where you were going. As I read through your general discussion, the value of this information became more apparent. This material might be more effective in your discussion than in your introduction.

7. I appreciated your explanation of why you had both a short exposure and long exposure conditions in your first study (pages 14-15). Being exploratory studies, having both conditions made excellent sense.

8. I was struck by the low alpha of your welding self-efficacy scale (.62) and switching to a shorter scale in the second study. Given the issues with this, and the fact that the measure is a central one in your research, you might want to present a factor analysis of the scale. You mention in your limitations (page 30) that the scale was not unidimensional. The factor analysis would be helpful. Do you think that part of the problem with this scale might have been your sample?

9. Why did you have two self-efficacy measures in the second study, one for the VR condition and one for the desktop condition? Given that the newer scale was longer (and so more likely to be reliable), why not just use it?

10. I was wondering about the difference in performance between the VR and desktop conditions (page 21). This may have occurred because the VR added a layer of difficulty (which you mention), but I wonder if the desktop condition added additional information. Were the other members of the group able to see the information on the desktop as other students performed? Was the feedback in the VR condition given via the VR or through the monitor? If the feedback in the VR condition was given via VR, other subjects in the desktop condition may have been better able to see the relationship between actions and performance. With the VR condition they would not have been able to see the feedback that the performer was getting from the system. This might also explain the stronger social learning effects with VR. If students in the desktop condition were also watching the feedback on the desktop, they might have overlooked some of the social cues from the learner who was practicing.

11. I thought that your effects on behavioral modeling (pages 22-23) were interesting. Can you describe how these compare with other examples of group learning? For example, has previous research with other tasks also demonstrated your finding that group members tended to imitate the learning strategy of others, regardless of their level of performance? Is the immediate learner more important than the first learner? You mention one study on page 27. Is this the only previous literature?

12. You make a good point on page 28 that, given your sample, your findings are probably a lower bound of effects. If you can find effects with this sample, they should be even larger with a sample more likely to be interested in welding as a career. The same is true with your point about how brief the experience was (page 29). Finding effects with such a short exposure suggest that these effects would be strong.

13. I appreciated your practical implications (pages 29-30). Your third limitation on page 31 (not having specific goals) could have been incorporated into the practical implications of your studies.

Reviewer #4: This manuscript is relevant to an eLearning trend that lies in the intersection between immersive educational applications and career development training. The RQs and hypotheses are well-grounded in theories.

One interesting finding of study 2 about the proximity effect within the group behavioral modeling could be more elaborated. This finding has some practical instructional design implications. For example, how to arrange group members to maximize the modeling outcomes? Another related question is the group observing procedure in Study 2 "To mimic a real group training environment (e.g., welding career and technical education), when not welding, group members were positioned

around the VR or desktop simulation to observe their peers on the monitor. " (p.18). I didn't know when and how each welder observes the 1st welder and the one before him/her. I suggest the authors clearly explain the procedure because the second paragraph in the Study 2 discussion (p.25-p.26) is all about the finding related to this point. But I couldn't see where those come from.

However, I wish the authors had included the interface images of all the conditions used in this study. Particularly, it would be much better for me to visualize the differences between the two EVEs (VR or desktop simulation) used in their studies.

6. PLOS authors have the option to publish the peer review history of their article (what does this mean?). If published, this will include your full peer review and any attached files.

Reviewer #1: No

Reviewer #2: **Yes: **Dr Steven V. Cates

Reviewer #3: No

Reviewer #4: No

---

## [Author Response · Author response to Decision Letter 0]

20 May 2022

Editor’s Comments

1. The reviewers expressed concerns about the focus of the study, clarification of the procedure as well as more critical discussions of the findings.

Authors’ Response: We appreciate the opportunity to address these concerns. We hope that our responses to the reviewer’s comments below and corresponding changes to the manuscript have ameliorated these concerns and strengthened the quality of the paper.

Reviewer 1 Major Comments

1. The paper is very ambitious, particularly given the fact that it is based on two studies. However, the present version of the paper demands a high degree concentration and focus or a significant cognitive load. This is because it is very difficult to grasp its key ideas and objectives. Most specifically, it was a challenging task keeping in mind the focus on both affordances of career-oriented EVE and the role that social learning play in facilitating learning affordances of the studies --- while trying to digest four research questions at the same time.

Authors’ Response: We thank the reviewer for the comment. We believe that the volume of information that our studies provide on the learning affordances of EVEs makes a significant contribution, but also understand that this could be cognitively taxing. One step we took to address this concern was restructuring parts of the introduction section (see p. 4). The introduction section now follows this structure: paragraph 3 presents the research gaps, paragraph 4 explains how our studies address these gaps, and paragraph 5 explains our use of an inductive approach. Previously, paragraphs 4 and 5 were reversed. We think that this structure improves the flow of the introduction and provide a clearer roadmap at the start of the paper of how each study addresses each research need. 

Another step we took to address this comment was to expand the "Overview of studies" section (pp. 12 - 13). We again provided clarity on the research gap addressed by each study and stated which research questions correspond to Study 1 and Study 2. Since this section immediately precedes the studies, we hope that it will reduce the cognitive load for the reader to understand the alignment between the research gaps, research questions, and studies. 

Lastly, in the results section for each study, we restated each research question. We believe that this will also reduce the cognitive load of having to recall the research questions and will serve as another indicator of the research purpose served by each study.

2. Study 2 appears to be strongest and the one that's better developed. Therefore, I recommend placing a stronger focus on it. Then, perhaps study 1 can be repurposed/repackaged as a conference paper.

Authors’ Response: We believe that removing Study 1 would substantially reduce the contribution of this paper. Study 1 may be simpler in its design than Study 2, but we think it addresses an important research need. Importantly, the two studies together give a complete picture of the phenomenon we hope to describe.

Reviewer 1 Other Adjustments Needed

1. Revision of the papers' abstract. It needs to be edited for clarity and readability. The verbiage used in some of the sentences is a bit unorthodox. Language used should be familiar, precise and unambiguous.

Authors’ Response: The abstract of the paper was revised to improve clarity and readability. Specifically, we attempted to more clearly define technical terms (e.g., immersion, fidelity, career-related attitudes) whose meaning may not be obvious to some readers. We also shortened and restructured some sentences to improve overall readability.

2. Inclusion of a demographics table. This will help the reader understand, at a glance, who the participants in the study(ies) are and gain a better perspective of their backgrounds. Some of that information is buried in the text. So, it’s hard to keep up with who is involved in the study.

Authors’ Response: We have added a demographics table to the manuscript that includes all of the background information that we collected on participants in both Study 1 and Study 2.

3. Break table 2 into small ones and place those where they are needed. Not sure all of the correlation figures are needed to be shown in one table. This would also make it possible to reduce or eliminate the amount of lookbacks to table 2 as the text is being read.

Authors’ Response: We elected not to implement this suggestion. Breaking up the correlation table would result in the omission of bivariate correlations between some of the study variables. 

Reviewer 2

1. This was an interssting study concerning two variables as it realtes to Self Efficacy and the usage of AI orm Visual Reality. the Research Deisgn was sound...The Research Questions came directly from the Literature. The statistical analysis was sound although some instruments created were not reliable for valid. Overall this was well done and impressive for a lead author who is un undergraduate student.

Authors’ Response: We thank the reviewer for this comment.

Reviewer 3 Major comments

1. My major concern with your paper (which the authors discuss) is that the learning being studied is welding training, but the samples for both studies are undergraduate psychology students. I think that the characteristics of the sample may have little impact on styles of learning and performance, but question what college undergraduates have to say about self-efficacy in a welding career. A sample from a community college or vocational school would have been much more useful. The sample may have also contributed to the testing issues with the career self-efficacy scale.

Authors’ Response: We understand the reviewer's concerns with the use of a student sample. We believe that a sample of welding trainees (or other trades) should be the ultimate goal for studies examining affordances of EVEs as a tool for improving career attitudes and performance in skills training. However, the purpose of the current study was to provide initial evidence of the affordances of EVEs for the outcomes of career exploration and skills training, and the effects of social learning dynamics in this context. Studies that establish phenomena empirically in a controlled setting are an important precursor to field research. Thus, a student sample was appropriate for our goals.

Further, the conclusions we draw do not go beyond the evidence. We are clear that the effects may be lower-bound estimates and that future research needs to determine whether our findings generalize (see "Practical implication" and "Limitations and future research directions" sections, pp. 32-33). Also, EVEs are not only useful for those already interested in or training for a job. As we discuss in the paper, EVEs can be used to allow students to explore careers that they would not have otherwise considered. Our findings are more directly applicable to such uses of EVEs, which we explain in the second paragraph of the "Practical implications" section (p. 32).

2. Another concern is that most readers would expect to find that as EVEs become more immersive (more virtual) they should have more positive effects. In the same way that a video will be better than a picture, we would expect virtual reality to be more effective than a desktop simulation, which should be more effective than watching a video. Can you make a stronger case for why your findings are important? For example, can your findings say anything about the CAMIL model you cite (Makransky & Peterson, 2021)?

Authors’ Response: Although our findings may seem intuitive from the perspective expressed by the reviewer, one can just as easily make the case that more immersive media should have negative effects on learning outcomes. For example, in the "Social learning and VR" section (p. 8-10), we present two competing possibilities explaining that higher levels of immersion and fidelity may have positive or negative effects on social learning. Namely, it is just as plausible to think that higher immersion and fidelity can distract learners from the task at hand and from the observation of others or increase their engagement with both. Additionally, the perspective presented by the reviewer assumes that the relationship between the immersive features of media and learning outcomes is linear (i.e., VR is better than video which is better than a picture). However, another possibility suggested by our findings is a nonlinear relationship such that greater immersion and fidelity is beneficial up to a certain point, but that there is a ceiling to this effect. We included a discussion of this possibility in the "General discussion" section (p. 30). The point we are making here is that the effects of immersion and fidelity are not obvious at all, and studies like ours are necessary to provide empirical evidence that support or refute theoretical arguments.

We do believe that our findings provide support for specific contentions of the CAMIL framework. Although the original manuscript made this point in the third paragraph of the "General discussion" section (p. 30), we agree that we should have provided more discussion on this point. In the revised manuscript, we have included more detail in this paragraph about how our findings address the CAMIL, and how future studies can expand on our study to address key contentions of the framework. 

Reviewer 3 More Specific Issues

1. I thought that the beginning of your introduction (pages 3-4) had a very good justification for studying EVEs.

Authors’ Response: We thank the reviewer for this comment.

2. You emphasize in the introduction that you are presenting inductive studies with exploratory research questions, Is this why you are not submitting it to a psychology journal? It is not theoretical enough?

Authors’ Response: No, we did not submit this paper to PLOS ONE due a lack of theory. We also disagree with the notion that psychology journals dismiss or minimize inductive research. Although psychological research tends to favor hypothetico-deductive approaches, inductive research is recognized as a complementary approach that is imperative for the discovery of new phenomenon, theory development, and challenging existing theory (Hambrick, 2007; Woo et al, 2017). Thus, just because a paper does not follow a hypothetico-deductive mold does not mean it is theoretically useless. To support this point, we cited an example of an exploratory study (published in a psychology journal) that demonstrates how such research can be used for theory development (p. 4). Similarly, we believe that the results of our study speak to existing theory (e.g., CAMIL; Makransky & Petersen, 2021) and have the potential to be useful in future theory development.

3. I thought that your discussion of immersion and fidelity on page 5 was overly theoretical and so not as clear as it could be. Is immersion focused on seeing and fidelity focused on touching?

Authors’ Response: We removed some of the theoretical elaboration about fidelity and added a more pragmatic explanation of immersion. It is not the case that immersion is seeing and fidelity is touching. Immersion is about an EVE providing a comprehensive experience via the engagement of senses and fidelity is the degree of realism of the simulated environment and/or task.

4. You mention that previous research has examined self-efficacy (Makransky et. al., 2019) but don’t describe this research. I assume it examined self-efficacy of the task, not career self-efficacy, but some discussion of it seems warranted.

Authors’ Response: We agree that this study (Makransky et al, 2019) deserves more explanation and have included additional information about its findings at the end of this paragraph (p. 4-5). The reviewer's assumption is correct that this study did not examine career self-efficacy but rather self-efficacy for engaging in laboratory safety protocols. Even if this study had examined career self-efficacy, our study would still make a contribution because they did not compare the association between self-efficacy and EVEs that differ in immersion and fidelity as we do in Study 1.

5. I wasn’t sure if the group of learners were together IN the EVE when you discussed the possible ways learners might model each other’s behaviors (pages 9-10). This was compounded in your methods section (page 18) where it wasn’t clear to me how the rest of the group observed the learner who was practicing. You need to make this procedure clearer. I think what happened is that each subject took a turn with the virtual goggles (or the monitor) to work on the welding simulation. Were the others physically with them while they practiced or did they watch through a monitor? After each subject practiced welding, did they observe the others, or did they fill out the questionnaire?

Authors’ Response: We appreciate the opportunity to clarify our procedure. The reviewer is correct that each participant in Study 2 took turns welding in the virtual environment. While an individual was welding, the other participants in the group were standing around them. For both conditions, group members were able to physically observe the participant who was welding and observe their weld through a monitor. Participants were told that they were welcome to provide feedback to one another. All participants took their turn welding and then all filled out the questionnaire at the end.

We revised the methods section of Study 2 to provide more clarity on the procedure (pp. 19-20). Additionally, we revised some of the language in the "Immersion, fidelity, and social learning dynamics" section (pp. 9-10) to make it clear in the literature review that we are interested in an EVE-based group skills training in which participants take turns using the EVE and others observe from the physical environment.

6. When I first read your discussion on Page 11, it wasn’t clear to me where you were going. As I read through your general discussion, the value of this information became more apparent. This material might be more effective in your discussion than in your introduction.

Authors’ Response: We agree with the reviewer that some of the information in the "Social learning, self-efficacy, and interest" section is not directly pertinent to the development of our fourth research question. In particular, the paragraph that discusses the contentious nature of the relationship between self-efficacy and performance including the role of time in this relationship and self-efficacy spirals is not directly relevant here. To address the issue, we took the reviewer's advice and moved it to the discussion section as a caveat with regard to the relationship between performance and self-efficacy. 

We do maintain that the other information in this section is necessary for setting up Research Question 4 because it discusses previous research on the relationship between performance and our career-related attitudes of interest (i.e., self-efficacy and interest) in other contexts. If we were to move this information to the discussion, we think the origin of Research Question 4 would become unclear. We hope that moving the paragraph mentioned above to the discussion clarifies how the rest of the section is relevant and how it flows from the previous section.

7. I appreciated your explanation of why you had both a short exposure and long exposure conditions in your first study (pages 14-15). Being exploratory studies, having both conditions made excellent sense.

Authors’ Response: We thank the reviewer for this comment.

8. I was struck by the low alpha of your welding self-efficacy scale (.62) and switching to a shorter scale in the second study. Given the issues with this, and the fact that the measure is a central one in your research, you might want to present a factor analysis of the scale. You mention in your limitations (page 30) that the scale was not unidimensional. The factor analysis would be helpful. Do you think that part of the problem with this scale might have been your sample?

Authors’ Response: To address the issues and questions surrounding the self-efficacy scale, we provided a supplementary document that details our process of selecting the self-efficacy items for both studies, including why we used different self-efficacy scales for the two conditions in Study 2 (see Response to Reviewer 3, Comment 9). In this document we also present the exploratory factor analysis showing the two-factor structure of the 4-item scale that we used in full in Study 1 and in part (two of the four items) for the VR condition in Study 2, and our interpretation of these two factors. We do not attribute the multidimensionality to the sample, but rather that self-efficacy is a very task-specific construct (Bandura, 2012). Because we used a different number of items from this scale for Study 1 and the VR condition of Study 2, and due to the exploratory nature of our research, we re-conducted the self-efficacy analyses from Study 1 and Study 2 for each of the two self-efficacy factors. We did find some differences in results between the two factors, and we provide a discussion of these differences.

9. Why did you have two self-efficacy measures in the second study, one for the VR condition and one for the desktop condition? Given that the newer scale was longer (and so more likely to be reliable), why not just use it?

Authors’ Response: We included a brief rationale in the "Measures" section (p. 21) for why we used different self-efficacy scales for the two conditions in Study 2, and also provided a more detailed explanation in the online supplement. In short, by the time we made the realization of the low reliability and multidimensionality of the 4-item self-efficacy scale, we had already used it for the first half of data collection for Study 2. Our goal in this study was not to compare the VR and desktop simulation conditions on career self-efficacy. Thus, we decided to switch to a measure that had been developed in past research to improve the internal validity of this part of the study. We expressly acknowledge in the Study 2 "Measures" section that any direct comparison of the VR and desktop simulation conditions on self-efficacy is inappropriate, and we do not do so in the manuscript.

10. I was wondering about the difference in performance between the VR and desktop conditions (page 21). This may have occurred because the VR added a layer of difficulty (which you mention), but I wonder if the desktop condition added additional information. Were the other members of the group able to see the information on the desktop as other students performed? Was the feedback in the VR condition given via the VR or through the monitor? If the feedback in the VR condition was given via VR, other subjects in the desktop condition may have been better able to see the relationship between actions and performance. With the VR condition they would not have been able to see the feedback that the performer was getting from the system. This might also explain the stronger social learning effects with VR. If students in the desktop condition were also watching the feedback on the desktop, they might have overlooked some of the social cues from the learner who was practicing.

Authors’ Response: The amount of information visible to group members was exactly the same in the VR and desktop conditions. For both conditions, a monitor was visible which presented the performance of the member who was welding in real time. The feedback screen that participants could view after each weld was also visible in the same manner for both conditions. We made some edits to the procedures section to make these points clearer to readers.

11. I thought that your effects on behavioral modeling (pages 22-23) were interesting. Can you describe how these compare with other examples of group learning? For example, has previous research with other tasks also demonstrated your finding that group members tended to imitate the learning strategy of others, regardless of their level of performance? Is the immediate learner more important than the first learner? You mention one study on page 27. Is this the only previous literature?

Authors’ Response: Some of our findings regarding social learning dynamics are in line with previous literature, while others have not yet been addressed to our knowledge. It is well-established that people do not engage in social learning indiscriminately, but rather tailor their social learning strategy to the context (Kendall et al, 2018). Relevant to the context of our study, people tend to rely on social information more when the learning context is uncertain (i.e., they have little prior information to bring to the task) (Toelch et al, 2014). The study we cited in the original manuscript also supports this notion. Given that our sample was inexperienced in welding and virtual environments, our strong social learning effects align with this past research. Importantly, this context of uncertainty does not explain the larger social learning effects that we found for the VR relative to the desktop simulation, which we attribute to the effect of immersion and fidelity. Additionally, we are unaware of any research that has examined the effects of ordering of behavioral models in social learning tasks. Thus, our finding that participants within-groups tend to copy more immediate behavioral models rather than the first behavioral model is novel.

In response to this comment, we included a greater description of how our findings relate to past research in line with what we presented above. We also moved this section from the "Study 2 discussion" section to the "General discussion" section, as the latter is where we discuss other parts of our findings in relation to existing research (e.g., CAMIL framework). We believe that this improved the flow of the paper and eliminated some redundancy between these two sections.

12. You make a good point on page 28 that, given your sample, your findings are probably a lower bound of effects. If you can find effects with this sample, they should be even larger with a sample more likely to be interested in welding as a career. The same is true with your point about how brief the experience was (page 29). Finding effects with such a short exposure suggest that these effects would be strong.

Authors’ Response: Yes, we agree with this point. We also point out that the low reliability of the self-efficacy scale means that the observed relationships between self-efficacy and other variables that we reported are attenuated relative to the true relationships.

13. I appreciated your practical implications (pages 29-30). Your third limitation on page 31 (not having specific goals) could have been incorporated into the practical implications of your studies.

Authors’ Response: We assume that the reviewer is referring to the practical implication of the importance of setting specific and challenging goals. While we agree with this point, we did not include it as a practical implication for two reasons. First, we did not collect data on participants’ goals and thus cannot draw any certain conclusions about their performance goals (or lack thereof). For example, participants could have developed their own performance goals in the absence of us providing them. We simply stated that our lack of providing goals is a possible explanation for some of our findings and a limitation that future research should rule out. Second, the research literature on the relationships between goal-setting and performance is robust (e.g., Locke & Latham, 2013). Thus, we do not feel that stating this point as an implication would contribute anything novel, especially because goals were not a focus of our study.

Reviewer 4 Major Comments

1. This manuscript is relevant to an eLearning trend that lies in the intersection between immersive educational applications and career development training. The RQs and hypotheses are well-grounded in theories.

Authors’ Response: We thank the reviewer for this comment.

2. One interesting finding of study 2 about the proximity effect within the group behavioral modeling could be more elaborated. This finding has some practical instructional design implications. For example, how to arrange group members to maximize the modeling outcomes?

Authors’ Response: We thank the reviewer for this suggestion. We added more detail on the practical implications of the proximity effects we found for instructional design (p. 31).

3. "To mimic a real group training environment (e.g., welding career and technical education), when not welding, group members were positioned around the VR or desktop simulation to observe their peers on the monitor. " (p.18). I didn't know when and how each welder observes the 1st welder and the one before him/her. I suggest the authors clearly explain the procedure because the second paragraph in the Study 2 discussion (p.25-p.26) is all about the finding related to this point. But I couldn't see where those come from.

Authors’ Response: We appreciate the opportunity to clarify our procedure. Reviewer 3 shared a similar concern which we addressed and responded to earlier (see response to Reviewer 3, Comment 5). We reiterate here how we addressed this concern. Each participant in Study 2 took turns welding in the virtual environment. While an individual was welding, the other participants in the group were standing around them. For both conditions, group members were able to physically observe the participant who was welding and observe their weld through a monitor. Participants were told that they were welcome to provide feedback to one another. All participants took their turn welding and then all filled out the questionnaire at the end.

We revised the methods section of Study 2 to provide more clarity on the procedure (pp. 19-20). Additionally, we revised some of the language in the "Immersion, fidelity, and social learning dynamics" section (pp. 9-10) to make it clear in the literature review that we are interested in an EVE-based group skills training in which participants take turns using the EVE and others observe from the physical environment.

4. However, I wish the authors had included the interface images of all the conditions used in this study. Particularly, it would be much better for me to visualize the differences between the two EVEs (VR or desktop simulation) used in their studies.

Authors’ Response: We agree with the reviewer that images would be helpful. Unfortunately, we are unable to provide images of the EVEs used in this study because of a confidentiality agreement with the company that provided us with these machines.

---

## [Decision Letter · Decision Letter 1]

21 Jun 2022

PONE-D-21-37956R1Social learning dynamics influence performance and career self-efficacy in career-oriented educational virtual environmentsPLOS ONE

Dear Dr. Pitcher,

Thank you for submitting your manuscript to PLOS ONE. After careful consideration, we feel that it has merit but does not fully meet PLOS ONE’s publication criteria as it currently stands. Therefore, we invite you to submit a revised version of the manuscript that addresses the points raised during the review process.

As pointed out by the reviewers, the significance of the study could be further highlighted and the justification of the chosen sample could be stronger too.

We look forward to receiving your revised manuscript.

Kind regards,

Mingming Zhou, Ph.D.

Section Editor

PLOS ONE

Journal Requirements:

Reviewers' comments:

Reviewer's Responses to Questions

**Comments to the Author**

1. If the authors have adequately addressed your comments raised in a previous round of review and you feel that this manuscript is now acceptable for publication, you may indicate that here to bypass the “Comments to the Author” section, enter your conflict of interest statement in the “Confidential to Editor” section, and submit your "Accept" recommendation.

Reviewer #3: (No Response)

Reviewer #4: All comments have been addressed

2. Is the manuscript technically sound, and do the data support the conclusions?

Reviewer #3: Yes

Reviewer #4: Yes

3. Has the statistical analysis been performed appropriately and rigorously? 

Reviewer #3: Yes

Reviewer #4: Yes

4. Have the authors made all data underlying the findings in their manuscript fully available?

Reviewer #3: Yes

Reviewer #4: No

5. Is the manuscript presented in an intelligible fashion and written in standard English?

Reviewer #3: Yes

Reviewer #4: Yes

6. Review Comments to the Author

Reviewer #3: Social learning dynamics influence performance and career self-efficacy in career-oriented 5 educational virtual environments (revised)

PLOS ONE

MS# PONE-D-21-37956 R1

Comments for the Author(s)

Major Issues:

My major concern with your paper was that the samples for both studies were undergraduate psychology students. You address this limitation in your discussion. I liked what you had to say in your response to me. I would suggest that you add this material to your general discussion. At this point you simply say that undergraduate psychology students are not ideal for studying this phenomenon. I think that you can make a stronger argument for why your sample is reasonable for this stage of the research.

The other concern I raised is that most readers would expect to find that as EVEs become more immersive (more virtual) they should have more positive effects. In your response to me, you point out that your introduction raises contradictory predictions for the impact of immersion. The fact that you can advance contradictory predictions and then show one was correct allows you to make a stronger case for why your findings are important. But you never refer back to these contradictory prediction in your discussions. Again, I think that you can make a stronger case for the value of your research by taking the points from the response that you made to me and including them in your general discussion.

More Specific Issues:

1. No changes necessary.

2. This comment was not intended to be a knock on Psychology journals or your research. I admired your inductive approach and recognized that it is different from most psychology journal studies. No changes necessary.

3. I thought that your discussion of immersion and fidelity were much clearer in this version. I found this to be much more helpful.

4. You now mention that Makransky et. al. (2019) studied self-efficacy of the task (and other attitudinal outcomes).

5. I thought that you did a much better job of describing how the learners interacted, both in the introduction (in the section on social learning dynamics) and in the procedure section of Study 2.

6. Your reorganization is much clearer (at least to me) with a shorter section in the introduction and moving one section to the discussion.

7. No changes necessary.

8. I like the notion of having the exploratory factor analysis and other information in an online supplementary document. This is important for those who may want to replicate or expand your research, but would tend to add bloat to your paper. You also may want to mention the supplementary document when you report the alpha in Study 1.

9. I appreciate your explanation in the welding self-efficacy section of Study 2 as to why you have two self-efficacy measures in the second study.

10. I was happy to learn that the information available to other subjects was the same in both conditions. Your new description of the procedures makes this clear.

11. I liked the changes you made concerning the behavioral modeling effects and continue to believe that these are very interesting and a real contribution in your research.

12. No changes necessary.

13. I didn’t see not having specific goals as a limitation since it would be difficult for new students of welding to employ specific goals (Faster welds? Better welds?) for their first attempts at welding.

Reviewer #4: This revision is much clearer and substantially improved. I can see where the RQs come from in the intro/lit review. The methods used to test the RQs were designed properly. Even though the discussion for the individual study is a little bit sweeping, the general discussion is well done in interpreting the findings of the two studies in a structured and practical way. Taking the last paragraph of study 2 discussion as an example (p.30), I would like to know if the literature talks about why only males' self-efficacy and interest were predicted by performance while you have 69% of female participants? The authors seem to explain it using the CAMIL framework in the general discussion (p.32), but to be honest, I still don't know how and why they are related. But overall, as a undergraduate's work, it is pretty good.

7. PLOS authors have the option to publish the peer review history of their article (what does this mean?). If published, this will include your full peer review and any attached files.

Reviewer #3: **Yes: **John L Cotton

Reviewer #4: No

---

## [Author Response · Author response to Decision Letter 1]

26 Jul 2022

Editor’s Comments

1. As pointed out by the reviewers, the significance of the study could be further highlighted and the justification of the chosen sample could be stronger too.

Authors’ Response: We appreciate the change to address the concerns raised by the reviewers. We hope that our responses below and corresponding changes to the manuscript have resolved these concerns and improved the paper.

Reviewer 3 Major comments

1. My major concern with your paper was that the samples for both studies were undergraduate psychology students. You address this limitation in your discussion. I liked what you had to say in your response to me. I would suggest that you add this material to your general discussion. At this point you simply say that undergraduate psychology students are not ideal for studying this phenomenon. I think that you can make a stronger argument for why your sample is reasonable for this stage of the research.

Authors’ Response: We took the language that we used in our response to the reviewer's initial comment and repurposed it for the "Limitations and future research" section to provide a stronger argument for our sample. We thank the reviewer for helping us refine our argument for the appropriateness of our sample.

2. The other concern I raised is that most readers would expect to find that as EVEs become more immersive (more virtual) they should have more positive effects. In your response to me, you point out that your introduction raises contradictory predictions for the impact of immersion. The fact that you can advance contradictory predictions and then show one was correct allows you to make a stronger case for why your findings are important. But you never refer back to these contradictory prediction in your discussions. Again, I think that you can make a stronger case for the value of your research by taking the points from the response that you made to me and including them in your general discussion.

Authors’ Response: We took the language that we used in our response to the reviewer's initial comment and repurposed it for the "Limitations and future research" section to provide a stronger argument for our sample. We thank the reviewer for helping us improve our communication of the significance of these findings.

Reviewer 3 More Specific Issues

1. No changes necessary.

Authors’ Response: No response.

2. This comment was not intended to be a knock on Psychology journals or your research. I admired your inductive approach and recognized that it is different from most psychology journal studies. No changes necessary.

Authors’ Response: We thank the reviewer for this comment.

3. I thought that your discussion of immersion and fidelity were much clearer in this version. I found this to be much more helpful.

Authors’ Response: We thank the reviewer for this comment.

4. You now mention that Makransky et. al. (2019) studied self-efficacy of the task (and other attitudinal outcomes).

Authors’ Response: No response.

5. I thought that you did a much better job of describing how the learners interacted, both in the introduction (in the section on social learning dynamics) and in the procedure section of Study 2.

Authors’ Response: We thank the reviewer for this comment.

6. Your reorganization is much clearer (at least to me) with a shorter section in the introduction and moving one section to the discussion.

Authors’ Response: We thank the reviewer for this comment.

7. No changes necessary.

Authors’ Response: No response.

8. I like the notion of having the exploratory factor analysis and other information in an online supplementary document. This is important for those who may want to replicate or expand your research, but would tend to add bloat to your paper. You also may want to mention the supplementary document when you report the alpha in Study 1.

Authors’ Response: We direct readers to the supplementary document in reporting alpha in Study 1.

9. I appreciate your explanation in the welding self-efficacy section of Study 2 as to why you have two self-efficacy measures in the second study.

Authors’ Response: We thank the reviewer for this comment.

10. I was happy to learn that the information available to other subjects was the same in both conditions. Your new description of the procedures makes this clear.

Authors’ Response: No response.

11. I liked the changes you made concerning the behavioral modeling effects and continue to believe that these are very interesting and a real contribution in your research.

Authors’ Response: We thank the reviewer for this comment.

12. No changes necessary.

Authors’ Response: No response.

13. I didn’t see not having specific goals as a limitation since it would be difficult for new students of welding to employ specific goals (Faster welds? Better welds?) for their first attempts at welding.

Authors’ Response: We agree with the point that it would have been difficult to provide specific performance goals to students with no prior familiarity with welding. Accordingly, we removed the corresponding paragraph in the "limitations and future research" section. We decided to remove the paragraph rather than move it to the “Practical implications” section as originally suggested by the reviewer because performance goals re not a focus of these studies. Removing the paragraph resulted in the elimination of one citation from the references list.

Reviewer 4

1. This revision is much clearer and substantially improved. I can see where the RQs come from in the intro/lit review. The methods used to test the RQs were designed properly. Even though the discussion for the individual study is a little bit sweeping, the general discussion is well done in interpreting the findings of the two studies in a structured and practical way. Taking the last paragraph of study 2 discussion as an example (p.30), I would like to know if the literature talks about why only males' self-efficacy and interest were predicted by performance while you have 69% of female participants? The authors seem to explain it using the CAMIL framework in the general discussion (p.32), but to be honest, I still don't know how and why they are related.

Authors’ response: We thank the reviewer for raising this point of clarification. We did not intend to explain the gender differences in the performance-career attitudes relationship with the CAMIL framework. Rather, we mentioned that these findings may reveal a boundary condition of the framework (p. 32, lines 711-713). Further, we are unaware of any literature that addresses why there may be a lack of a relationship between performance and career attitudes for women in the context of VR which is why we call for future research to investigate this finding in greater depth (p. 33).

---

## [Decision Letter · Decision Letter 2]

16 Aug 2022

Social learning dynamics influence performance and career self-efficacy in career-oriented educational virtual environments

PONE-D-21-37956R2

Dear Dr. Pitcher,

We’re pleased to inform you that your manuscript has been judged scientifically suitable for publication and will be formally accepted for publication once it meets all outstanding technical requirements.

Kind regards,

Mingming Zhou, Ph.D.

Section Editor

PLOS ONE

Additional Editor Comments (optional):

Reviewers' comments:

Reviewer's Responses to Questions

**Comments to the Author**

1. If the authors have adequately addressed your comments raised in a previous round of review and you feel that this manuscript is now acceptable for publication, you may indicate that here to bypass the “Comments to the Author” section, enter your conflict of interest statement in the “Confidential to Editor” section, and submit your "Accept" recommendation.

Reviewer #3: All comments have been addressed

2. Is the manuscript technically sound, and do the data support the conclusions?

Reviewer #3: Yes

3. Has the statistical analysis been performed appropriately and rigorously? 

Reviewer #3: Yes

4. Have the authors made all data underlying the findings in their manuscript fully available?

Reviewer #3: Yes

5. Is the manuscript presented in an intelligible fashion and written in standard English?

Reviewer #3: Yes

6. Review Comments to the Author

Reviewer #3: (No Response)

7. PLOS authors have the option to publish the peer review history of their article (what does this mean?). If published, this will include your full peer review and any attached files.

Reviewer #3: **Yes: **John L Cotton

---

## [Editor Report · Acceptance letter]

5 Sep 2022

PONE-D-21-37956R2 

Social learning dynamics influence performance and career self-efficacy in career-oriented educational virtual environments 

Dear Dr. Pitcher:

I'm pleased to inform you that your manuscript has been deemed suitable for publication in PLOS ONE. Congratulations! Your manuscript is now with our production department. 

Kind regards, 

on behalf of

Dr. Mingming Zhou 

Section Editor

PLOS ONE